# Maternal occupation and risk of adverse fetal outcomes in Tanzania: A hospital-based cross-sectional study

Baldwina Tita Olirk[1]*, Aiwerasia Vera Ngowi[2], Furaha August[3], Ezra Jonathan Mrema[2], Jovine Bachwenkizi[2], Simon Henry Mamuya[2]

1 Department of Occupational Health and Safety, Muhimbili National Hospital, Dar es Salaam, Tanzania, 2 Department of Environmental and Occupational Health, Muhimbili University of Health and Allied Sciences, Dar es Salaam, Tanzania, 3 Department of Obstetrics and Gynecology, School of Medicine, Muhimbili University of Health and Allied Sciences, Dar es Salaam, Tanzania

* baldwina.olirk@mnh.or.tz

## Abstract

### Background

Women constitute a large proportion of the workforce in today's world. Hazardous working environment conditions for these women pose threat to their reproductive health. Despite efforts to address maternal health in Tanzania, the impact of occupational risks during pregnancy remains unclear. We assessed whether maternal occupation during pregnancy is associated with adverse Foetal outcomes.

### Methods

A cross-sectional study was conducted among 400 self-referred post-delivery women at a referral Hospital in Tanzania. Information on socio-demographic characteristics and maternal occupational characteristics was assessed through the use of a pre-tested questionnaire. Questions on physical demanding work and prolonged standing were obtained from the standardized Musculoskeletal Questionnaire. To assess occupational exposure to chemicals, job titles and task descriptions were linked to a job-exposure-matrix, an expert judgment on exposure to chemicals at the workplace. Information relating to obstetric characteristics and pregnancy outcomes was obtained from the medical files and clinic cards. Data was analyzed by using Statistical Package for Social Sciences (SPSS) version 23. Odds ratios > 1 was considered risk while Odds ratios < 1 was considered protective and P value < 0.05 was considered significant.

### Results

The mean age was 28.0 ± 6.3. Out of 400 post-delivery women studied, 174 (43.5%) were engaged in various occupations. Agriculture (22.4%) was the most prevalent occupation followed by tailoring (19.0%). Relative to the referent group of other occupations, agriculture workers, had higher adjusted odds ratios of congenital malformation (AOR = 4.5, 95% CI; 1.6-12.8)preterm babies (AOR = 2.8, 95% CI; 1.3-7.9), low birth weight (AOR = 3.1, 95%

**Data availability statement:** All relevant data are within the paper and its Supporting Information files.

**Funding:** Funding was provided by SIDA project from Muhimbili University of Health and Allied Sciences. The funders had no role in study design, data collection and analysis, decision to publish, or preparation of the manuscript.

**Competing interests:** The authors have declared that no competing interests exist.

CI; 1.4-8.4) and low Apgar score (AOR = 3.5, 95% CI; 1.3-9.5). Food vendors: low birth weight (AOR = 8.6, 95% CI; 2.7-24.8) and low Apgar score (AOR = 13.5, 95% CI; 4.5-39.4).

## Conclusion

Understanding occupational characteristics and their relation to adverse Foetal outcomes is important to formulate appropriate strategies to promote and protect maternal and infant health at work.

## Introduction

Adverse Foetal outcome has been a worldwide threat and remains to be a major cause of perinatal morbidity and mortality [1–3]. Globally, it was reported that 2·42 million neonates died in 2019 with the majority of death occurring in Sub-Saharan Africa and South Asia [4]. During an assessment of under-five mortality rate, it was seen that the rate in Sub-Saharan Africa and South Asia increased from 73% (7·07 million deaths) in 2000 to 80% (4·04 million deaths) in 2019 [4]. Risk factors for adverse Foetal outcomes include maternal age, body mass index, obstetric factors and pregnancy-related syndromes including hypertensive disorders in pregnancy and other medical co-morbidities like HIV, Malaria, Anemia and sickle cell disease [5–7].

Women constitute almost half of the workforce globally, majority work during their reproductive age [8]. In England, the USA and Iran, 70%, 59% and 13% of women are employed respectively [9]. In Tanzania, according to the World Bank collection of development indicators report of 2020, female labor force, accounts for 48.13% [10]. This group include pregnant women and women of reproductive age. A large portion of these women work in informal sectors and hazardous occupational conditions including long periods of standing, long periods of walking, lifting or carrying heavy weights, night shifts, long working hours, exposure to chemicals such as pesticides, phthalates, organic solvents, alkylphenolic compounds, heavy metals and other chemicals [11]. The exposure to such hazardous occupational conditions may affect negatively the Foetal outcome. An analysis of 13 European birth cohorts that evaluated the influence of maternal occupation on birth weight, and length of gestation revealed higher odds of low birth weight (LBW) babies and longer length of gestation for the women who worked in construction industry [8]. Likewise, it was observed that women who worked in food industry had higher odds of preterm delivery and reduced length of gestation [8].

Additional studies on occupational exposures such as chemicals and noise has been related with reproductive defects in women and effects in pregnancy [11,12]. Numerous studies have acknowledged that occupational exposure to chemicals and toxins to women working in different occupations such as pesticides from agriculture, chemicals from salons, health centers and heavy metals can lead to teratogenic effect to their unborn babies [11,13]. A study that was done in Florida among female agricultural workers while pregnant revealed that, female working in agriculture sector reported more health symptoms during their pregnancy than female workers in a control group [14]. In addition, studies on chemical exposure, a study among cosmetologists and manicurists in California revealed that women who work in these industries during pregnancy are at higher risks for small for gestational age (SGA) compared to other working women and general population [15].

Maternal employment during pregnancy in relation to Foetal outcome has been rarely studied and most of them have generated contradictory findings [16]. On the other hand, the women in labor force is actually highest in the sub-saharan African compared to any other regions globally. In Sub-Saharan Africa, the labor force participation rate in females is 60.7%

compared to 50% of the global rate [17]. It is still not clear on how women in Tanzania should be handled during pregnancy. The contradictory findings regarding maternal occupational health risks, emphasize the need for further investigation in this area. This study assessed whether maternal occupation during pregnancy is associated with adverse Foetal outcomes. The results broaden knowledge about health of pregnant women and their fetus. The findings are also useful to researchers conducting future research, and for policy makers reviewing the policies to include issues that address how women should be handled during the maternal period.

## Materials and methods

### Study design

This analytical cross-sectional study was conducted to determine whether maternal occupation during pregnancy is associated with adverse Foetal outcomes.

### Setting

The Study was conducted at Muhimbili National Hospital (MNH). MNH is the largest referral hospital in Tanzania and a teaching hospital for Muhimbili University of Health and Allied Sciences. The Hospital is located in Dar-es-Salaam city with a population of 5.4million (Tanzania Demographic Census 2022). It serves populations of different levels of education and socioeconomic status. It also attends referred patients from different regions and so caters to a large number of wide ranges of patients including those with diverse occupations and pregnancy outcomes. MNH also harbors many specialists who are capable of making correct diagnosis. It is the largest referral hospital in the United Republic of Tanzania with 1,500 bed facility, attending 1,000 to 1,200 outpatients per day, admitting 1,000 to 1,200 inpatients per week. The hospital receives maternal cases mostly from the regional hospitals and very few from outside the country. The hospital has 2 maternity blocks in which women with pregnancies or having delivered pregnancies of 28 weeks or more are admitted. There are seven wards, four wards are reserved for admission of women with antenatal and postnatal complications and women with sick children. The number of deliveries per day is between 20 and 40. Also, there are two gynecological wards, each with 32 beds. The gynecological wards admit women with pregnancies of less than 28 weeks of gestation who have developed complications together with non-pregnant women with gynecological conditions. Emergency operations are conducted either in the Emergency medicine operation theatre or a gynecological operating room at the main theatre. The site was purposively selected with consideration of high number of deliveries.

### Study subjects and sampling methodology

The study was conducted from April to June 2022. The method used for sample size estimation was probability sample size estimation developed by Bhopal [18]. We were able to obtain a sample size of 400 with a response rate of 80%. Purposive sampling was conducted including working women (paid labour and self-employed) and non-working women within 72 h of delivery. An inclusion criterion was singleton pregnancy leading to live birth to minimize variability and reduce confounding factors as multiple pregnancies may present different physiological responses and outcomes.[19]. The exclusion criteria were maternal cormobities including cardiac, renal disease, sickle cell and HIV. The diagnosis of these cormobities was based on documented medical records provided by the Hospital. To account for the design effect, the sample size was multiplied by 1.5.

## Data collection procedure and tools

**Exposure.** A combination of occupation (job title) evidenced to be performed by women in Tanzania was considered for all jobs held by mothers. Jobs held from three months before pregnancy to the end of the third pregnancy month were considered exposures. A team of two occupational experts collected information on mothers' employment status and occupational history. Up to eight jobs held during pregnancy and 3 months before pregnancy were recorded and coded for each woman. However, only jobs held during the critical period of Foetal development, defined as 1 month prior to pregnancy through the end of the third pregnancy month, were included in these analyses. Questions on physical demanding work, manually handling loads of 25 kg and above, prolonged standing, night shifts, and working hours were obtained from the standardized Musculoskeletal Questionnaire. Information on work schedules included the nature of shift (either fixed verses rotating shift) and night shift. Information on working hours was calculated by number of working hours per day in the category of < 8 hours and > 8 hours. Mothers were also asked to rank the weight of their tasks into (light, normal, heavy, very heavy). Mothers were also asked to estimate the length of experience (in years) for the current job. To assess occupational exposure to chemicals, job titles and task descriptions were linked to a job-exposure-matrix (JEM), an expert judgment on exposure to chemicals at the workplace.

A face-to-face interview with delivering mothers was conducted using a structured questionnaire. Together with the collection of occupation related information described above, the questionnaire also gathered information on socio-demographic information (such as age, educational status, marital status). The questionnaire was written in English and translated into Swahili.

**Outcome variable.** The outcome variable was adverse Foetal outcome. These included low birth weight, early neonatal death, low APGAR scores, prematurity, stillbirths and physical congenital anomalies. Low birth weight was defined as birth weight of less than 2500grams, prematurity as neonates born at less than 37 weeks gestational age, still birth as the baby with no sign of life at or after 28 weeks of gestational age, low Apgar score as a score below 7. This information was obtained from the medical records and transferred to a structured checklist with the help of trained midwives. Addition to foetal information, obstetric data was also collected including gravidity, parity, gestational age at booking and other maternal complication such as eclampsia, preeclampsia, gestational diabetes, chronic hypertension, APH and uterine rupture.

## Data analysis

The dependent variable, adverse Foetal outcome (i.e., low birth weight, early neonatal death, low APGAR scores, prematurity, stillbirths and physical congenital anomalies), were measured as categorical variables where 1 was the presence of any adverse Foetal outcome and 2 was the absence of adverse Foetal outcome. The prevalence of adverse Foetal outcome was calculated by the formula developed by Bhopal [18].

## Statistical analysis

The data was analyzed by using IBM Statistical Package for Social Sciences (SPSS) for Windows version 23 (IBM Corp., Armonk, New York, USA) applying both descriptive and inferential statistical approaches. Inferential statistical approaches applied were chi-square test and logistic regression where p-value less than 0.05 was considered statistically significant. Odds ratios > 1 was considered as risk while Odds ratios < 1 was considered protective. Chi-square test was used to show the relationship between adverse Foetal outcome

and independent variables. Logistic regression was used to determine factors which affected the outcome variable independently while correcting for confounders (age, BMI, obstetric factors).

### Ethics approval and consent to participate

All methods were carried out in accordance with Declaration of Helsink guidelines and regulations. Ethics approval was obtained from ethics committee of Muhimbili University of Health and Allied Sciences with reference number DA.282/298/01.C/938. The permission to conduct the study at the referral hospital was obtained from the Executive Director (ED). Participants 18 years and above were given written informed consent forms to consent to participate in the study while parental/guardian consent was obtained for participants less than 18 years. Although we had 14 participants who had no formal education, they could read and write. The study participants who were found to have adverse Foetal outcome were given appropriate care.

## Results

### Background characteristics of study participants

We contacted 400 post-delivery women resulting in a participation rate of 70%. The mean age, gravidity, parity, and ANC visits of the study population were 28.0 ± 6.3, 3 ± 2, 2 ± 1, and 4.5 ± 1.9, respectively. Almost half of study participants had primary education (47.5%) and majority being married (74.2%). The rest of the characteristics are shown in Table 1

### Prevalence of adverse fetal outcomes

Among 174 (43.5%) working women, 153 (38.25%) had at least one adverse Foetal outcome. The commonest adverse outcomes in working women were low birth weight (27.1%) followed by preterm babies (24.5%). Table 2 shows distribution of risk for adverse Foetal outcomes between women who were working and those who were not working during pregnancy.

### Distribution of risk factors

**Occupational status.** In our study, post-delivery women self-reported to engage in various jobs during pregnancy (Fig 1). Out of 400 post-delivery women studied, 174 (43.5%) were engaged in various occupations. The most prevalent occupation was agriculture (22.4%) followed by tailoring (19.0%). The distribution of various Foetal outcomes across different occupational categories is presented in Table 3.

**Relationships between adverse fetal outcome and maternal occupation.** Working in agriculture sector during pregnant had higher adjusted odds of congenital malformation (AOR = 6.3, 95% CI; 1.4-28.8) preterm babies (AOR = 3.9, 95% CI; 1.1-15.8), low birth weight (AOR = 4.1, 95% CI; 1.2-16.5) and low Apgar score at 5minutes (AOR = 7.4, 95% CI; 2.0-30.0). The rest of the variables are shown in Table 4.

## Discussion

In this study, we assessed maternal occupations associated with adverse Foetal outcome among post-delivery women in Tanzania. The commonest types of occupations were agriculture, tailoring and hairdressing. This finding is not surprising as majority of women in Tanzania are engaged mainly in informal sectors like Agriculture and Hairdressing [20,21].

In this study, 38.25% of the working women experienced at least one adverse Foetal outcome. This prevalence is higher compared to the 29.7% prevalence of adverse birth outcomes in

**Table 1. General characteristics of study participants (N = 400).**

| Character | Frequency (n) | Percentage (%) |
|---|---|---|
| **Age (years)** | | |
| <17 | 1 | 1% |
| 17-25 | 146 | 36% |
| 262-35 | 201 | 50% |
| >35 | 52 | 13% |
| **Education level** | | |
| No formal education | 14 | 3.5% |
| Primary education | 109 | 47.5% |
| Secondary education | 133 | 33.3% |
| Higher education | 63 | 15.8% |
| **Marital status** | | |
| Single | 103 | 25.8% |
| Married | 297 | 74.2% |
| **History of diabetes** | | |
| Yes | 18 | 4.5% |
| No | 382 | 95.5% |
| **History of hypertension** | | |
| Yes | 36 | 9.0% |
| No | 364 | 91.0% |
| **Alcohol use during Pregnancy** | | |
| Yes | 22 | 5.5% |
| No | 378 | 94.5% |

Sub-Saharan Africa [22]. The higher prevalence could be attributed by occupational exposures and lack of occupational health services especially in the informal sectors [12]. The nature of employment among the majority of women, who are engaged in high-risk and informal occupations such as agriculture, significantly increases the risk to the unborn child. These occupations often involve hazardous working conditions including physical demanding work, limited break, exposure to harmful chemicals, and limited workplace protections [20]. Tanzania is witnessing rapid growth of informal sector particularly agriculture, accounting from 65 and 70 percent of this sector in Sub-Saharan Africa [23]. Occupational health and safety services in Tanzania governed by a government agency (The Occupational Safety and Health Authority), are mainly accessible to workers in the formal sectors such as large-scale mining and factories [12]. However, even in the formal workplaces, only about 5% have access to occupational health services [6]. In addition, the Authority falls short in adequately addressing gender dimensions concerning occupational health issues [24].

In this study, the most prevalent adverse outcomes in working women were low birth weight (27.1%) and preterm babies (24.5%). The prevalence of low birth weight is higher compared to the study done in Iran where the prevalence of low birth weight in working women was 6% [24]. These differences could be attributed to variations in how the exposures were assessed. For example, in the Iranian study, exposure assessment based on occupational characteristics such as work shift, posture and physical demanding work, leaving out the type of occupation and therefore not capturing the same range of occupational hazards. In our study, we assessed type of occupation. Women engaging in occupations such as agriculture and street vending may experience specific occupational risks that contribute to low birth weight including hazardous chemicals [25] and other environmental pollutants [26] and therefore increasing chances of obtaining higher prevalence of low birth weight. However, the prevalence of

Table 2.  Distribution of adverse fetal outcome among working and non-working women.

| Neonatal Outcome | Category | | P value |
|---|---|---|---|
| | Working women | Non-working women | |
| **Presence of any adverse outcome** | | | |
| Yes | 153 (87.9%) | 162 (71.7%) | <0.0001 |
| No | 21 (12.1%) | 64 (28.3%) | |
| **Low Apgar score at 5 minutes** | | | |
| Yes | 71 (40.8%) | 45 (19.9%) | <0.0001 |
| No | 103 (59.2%) | 181 (80.1%) | |
| **Still birth** | | | |
| Yes | 15 (8.6%) | 7 (3.1%) | 0.016 |
| No | 159 (91.4%) | 219 (96.9%) | |
| **Low birth weight** | | | |
| Yes | 108 (62.1%) | 66 (37.9%) | <0.0001 |
| No | 83 (36.7%) | 209 (52.3%) | |
| **Preterm babies** | | | |
| Yes | 98 (56.3%) | 72 (31.9%) | <0.0001 |
| No | 76 (43.7%) | 154 (68.1%) | |
| **Early neonatal death** | | | |
| Yes | 5 (2.9%) | 12 (5.3%) | 0.231 |
| No | 169 (97.1%) | 214 (94.7%) | |
| **Congenital anomaly** | | | |
| Yes | 55 (31.6%) | 43 (19.0%) | 0.004 |
| No | 119 (68.4%) | 183 (81.0%) | |

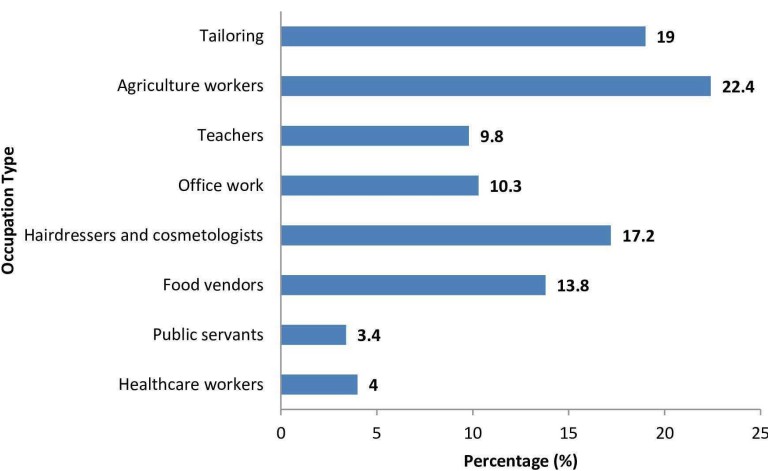

**Fig 1.  Job titles for study participants.**

low birth weight among working women in this study is similar to the findings from a study in Japan, which incorporated data from 1995 to 2015, and reported a 26.1% prevalence of low birth weight among working women in 2015 [27]. Unlike the study in Iran, the Japanese study accounted for a wide range of occupations, including agriculture, forestry, and fisheries, which may account for the similarities in findings with our study.

Table 3. Distribution of adverse fetal outcomes across occupational categories.

| Occupational Category | Foetal Outcome | P-Value |
|---|---|---|
| | **Congenital malformation** | <0.0001 |
| Health Care workers | 1 (14.3%) | |
| Food Vendors | 2 (8.3%) | |
| Hairdressers and cosmetologists | 7 (23.3%) | |
| Office work | 3 (16.7%) | |
| Teachers | 7 (41.2%) | |
| Agriculture workers | 26 (66.7%) | |
| Tailors | 9 (27.3%) | |
| | **Preterm babies** | |
| Health Care workers | 3 (42.9%) | 0.004 |
| Food Vendors | 12 (50%) | |
| Hairdressers and cosmetologists | 12 (40%) | |
| Office work | 13 (72.2%) | |
| Teachers | 9 (52.9%) | |
| Agriculture workers | 30 (76.9%) | |
| Tailors | 19 (57.6%) | |
| | **Low birth weight** | |
| Health Care workers | 4 (57.1%) | 0.002 |
| Food Vendors | 19 (79.2%) | |
| Hairdressers and cosmetologists | 15 (50%) | |
| Office work | 13 (72.2%) | |
| Teachers | 12 (70.6%) | |
| Agriculture workers | 29 (74.4%) | |
| Tailors | 16 (48.5%) | |
| | **Low Apgar score at 5 minutes** | |
| Health Care workers | 4 (57.1%) | 0.002 |
| Public Servants | 3 (50%) | |
| Food Vendors | 20 (83.3%) | |
| Hairdressers and cosmetologists | 10 (33.3%) | |
| Office work | 10 (55.6%) | |
| Teachers | 6 (35.3%) | |
| Agriculture workers | 22 (56.4%) | |
| Tailors | 9 (27.3%) | |

The prevalence of preterm birth is higher compared to the study done in Brazil where the prevalence of preterm birth in working women was 16.2% [28]. There could be various reasons for this diffrences including comprehensie prenatal health care services provided in Brazil and socioeconomic disparities faced by women working in low and middle income countries. Women working in the informal sectors in low and middle income countries face financial hardship [29] which could lead to poor nutrition and therefore affecting pregnancy. The types of occupation positively associated with the largest number of adverse Foetal outcomes were agriculture. Working in agricultural sector was positively associated with congenital malformation, low birth weight, low Apgar score and preterm babies. The higher odds of adverse Foetal outcomes among agricultural female workers are related to pesticide exposure [25,30]. However, higher prevalences are reported among agriculture workers in this study compared to studies done in well resource settings. For example, the prevalence of preterm

**Table 4. Association of maternal occupation and fetal outcomes.**

| Congenital malformation | p value | COR (95% CI) | P value | AOR (95% CI) |
|---|---|---|---|---|
| **Occupation type/characteristics** | | | | |
| Food vendors | 0.090 | 0.2 (0.05-1.2) | 0.075 | 0.2 (0.04-1.1) |
| Working in Agriculture | 0.001 | 5.3 (1.9-14.7) | 0.004 | 4.5 (1.6-12.8) |
| **Preterm babies** | | | | |
| **Occupation type/characteristics** | | | | |
| Working in Agriculture | 0.083 | 2.5 (0.9-6.8) | 0.045 | 2.8 (1.3-7.9) |
| **Low birth weight** | | | | |
| **Occupation type/characteristics** | | | | |
| Food vendors | 0.022 | 4.0 (1.2-13.2) | 0.011 | 8.6 (2.7-24.8) |
| Working in Agriculture | 0.026 | 3.1 (1.2-8.3) | 0.028 | 3.1 (1.4-8.4) |
| **Low Apgar score at 5 minutes** | | | | |
| **Occupation type/characteristics** | | | | |
| Working in Agriculture | 0.015 | 3.5 (1.3-9.3) | 0.014 | 3.5 (1.3-9.5) |
| Food vendors | <0.001 | 13.3 (3.6-46.8) | <0.001 | 13.5 (4.5-39.4) |

Controlled for age, gestation hypertension, gestation diabetes, pre eclampsia and eclampsia.

babies among women in agricultural occupations in this study is 76.9%, significantly higher than the 24.6% reported in a study from Japan which analyzed data from 2007 to 2019, specifically reported this prevalence for women in agricultural occupations in 2019 [31]. Despite agricultural occupation is hazardous, higher prevalence of preterm babies could be due to differences in working conditions. Farming and Agriculture in Tanzania may not be as mechanized as in the Japan, California and other well resource Countries. In addition, Farming and Agriculture still remain the commonest occupation in most of sub–Saharan Africa [20].

Women agricultural workers constitute a key occupational group of workers who experience pesticide exposure at childbearing age [20]. They experience indirect exposure through harvesting, planting, and soil preparation, re-entry activities such as gathering and bunching flowers, clean-up activities like washing contaminated clothes of their husbands and handling/ reusing pesticides empty containers. Epidemiologic and animal studies have acknowledged that occupational exposure to chemicals and toxins to women working in different occupations such as pesticides from agriculture, can lead to adverse effect to their unborn babies [32,33].

These effects have also been pronounced elsewhere. For example, a case–control study by Addissie et al. examined whether maternal exposure to pesticides increased the risk of birth defects [30]. A high prevalence of congenital malformation at birth was revealed among cases than controls. Harley et al. conducted a study to evaluate whether maternal pesticide exposure had effect on the new born [34]. The study showed decreased foetal growth and length of gestation in new-borns whose mothers were exposed to pesticides during pregnancy. Ling et al in 2018 studied prenatal exposure to pesticides in connection with preterm birth and low birth weight and found a strong association of preterm birth with pesticide exposure [25]. In addition, ecological and cross-sectional studies have conveyed positive associations between adverse foetal outcome (particularly preterm birth and low birth weight) and prenatal pesticide exposure [35,36]. Small biomarker-based studies with measured pesticides and their metabolites in maternal blood, urine, or umbilical cord blood have reported positive associations with the adverse foetal outcome particularly preterm birth and low birth weight [37,38].

The proportion of Preterm birth (50%), and low birth weight (79.2%) are higher compared to those reported in Ghana where the proportion of preterm birth and low birth weight was 25.7% and 23.8% respectively [39]. These differences could be due to environmental factors. In Tanzania, environmental pollution including traffic emissions, poor air quality has been documented [26].

Mothers who were food vendors were at higher risk of delivering newborns with low birth weight and low Apgar score. This finding is supported by other studies. Amegah and colleagues found an association of low birth weight among street food vendors [39]. This association might be due to automobile air pollution and wind-blown dust emission from the roads [26]. In Tanzania high levels of particulate matter ranging from 11-20 µg/m3 have been reported since 2005 [26]. Studies in well resource countries have documented pregnancy outcome and air pollution by assessing proximity of the residents to the roads. For instance, a study in Japan by Yorifuji et al reported residents within $\leq 50$ m to be 1.5 times of getting newborn with low birth weight [40]. According to a thorough analysis of the data on ambient air pollution by Stieb et al, it was revealed that there was a 11.4 g decrease in birth weight per 1 ppm CO, 28.1 g decrease per 20 ppb $NO_2$ and 16.8 g per $20 g/m^3$ PM10 [41]. A study involving fifteen countries in Africa showed significant associations between prenatal exposure to particulate matter and increased odds of LBW [26].

Our study has several notable strengths. One of these strengths is that this study is among the few studies in Tanzania that assessed the effects of maternal occupation and birth outcomes addressing a substantial existing gap in maternal and child health research. Second, the robust methods that were applied particularly the use of a standardize questionnaire for assessing ergonomic exposure and a job exposure matrix for assessing chemical exposure ensured validity and reliability in the data collection. Third, the Focus on a variety range of occupations provided valuable insights of potential occupational risk that may affect the health of women and fetus in resource limiting areas for appropriated future interventions.

This study had several limitations. First, our study employed purposive sampling which could limits the generalizability of our findings due to potential selection bias. Second, recall and reporting bias may affect the accuracy of self-reported data, particularly regarding occupational histories. Participants may have difficulty accurately recalling past exposures or experiences, which could lead to underreporting or misreporting of certain factors. Additionally, reporting bias could arise if participants were reluctant to disclose certain information, especially in sensitive areas such as pesticide use or health conditions, potentially skewing the data. Third, confounding variables such as socioeconomic factors (e.g., income, education level) and access to healthcare can influence both the type of occupation and health outcomes, which may create spurious associations between occupational exposures and adverse health outcomes if not properly controlled. For instance, individuals with higher socioeconomic status may have better access to healthcare, which could mitigate the effects of occupational exposures. Although an 80% response rate is reasonable, non-response bias due to the 20% non-response rate and the unavailability of data to assess this bias may introduce uncertainty regarding the generalizability of our findings. Data collection exclusively within 72 hours of delivery excluded women giving birth outside the Hospital (study site), this potentially introduces systematic bias by underrepresenting specific populations, such as those in rural or underserved areas. Last, most women in sub Saharan African are multitasking to keep their home. For example, most of the people documented as working in agriculture sector could still be street food vendors. This study didn't assess the cumulative effect of this.

## Conclusion

This study contains findings from analyses on a variety of jobs and Foetal outcome in order to generate hypotheses. It also provides information that will help direct future research

in relation to adverse Foetal outcome and workplace hazards. At least one adverse Foetal outcome was seen in 38.25% of the working women. The commonest adverse outcomes in working women were low birth weight (27.1%) followed by preterm babies (24.5%). The type of occupation/job positively associated with several adverse Foetal outcomes included agriculture and street food vendors. These findings point to the necessity of exercising caution while scheduling job obligations during pregnancy and carefully monitoring the fetus' growth during prenatal appointments.

Future research should; employ random sampling to enhance external validity and validate these findings across broader populations, employ objective monitoring of occupational exposure levels to complement self-reported data, expand the data collection timeframe including home births to capture a more representative sample, incorporate follow-up assessments at key intervals for example six months or one year postpartum to better understand the extended trajectory of neonatal and maternal health and assess cumulative effects of the multi-tasks particularly among women in Sub-Saharan countries

To safe guard health of pregnant women in the work place, several interventions are necessary. These includes first, establishing workplace policies that protect pregnant workers, such as limiting exposure to hazardous chemicals and physically demanding tasks, provide mandatory training on safe work practices for pregnant employees and ensure they have the option for alternative duties or leave during pregnancy, whenever required. Second, for women working in occupations involving chemical hazards such as agriculture, tailored interventions like providing protective equipment to minimize exposure to chemicals and other hazards are essential. Third, policymakers should come up with compelling regulations that target protection of pregnant women in high risk occupations especially in the informal sectors such as agriculture, hair dressing and tailoring. Lastly, integrating occupational health risk assessment in the routine antenatal care follow-ups will help early identification of risks for timely management to protect both the pregnant women and their fetus.

## Supporting information

**S1 Data. The data set used for analysis.**
(XLSX)

**S1 File. This is the questionnaire in Kiswahili.**
(DOCX)

**S2 File. This is the questionnaire in English.**
(DOCX)

**S3 File. This is the observation checklist** .
(DOCX)

## Acknowledgement

We would like to thank post-delivery mothers who spared their precious time to participate in our study.

## Author contributions

**Conceptualization:** Baldwina Tita Olirk, Aiwerasia Vera Ngowi.

**Data curation:** Baldwina Tita Olirk, Simon Henry Mamuya.

**Formal analysis:** Baldwina Tita Olirk, Jovine Bachwenkizi, Simon Henry Mamuya.

**Funding acquisition:** Baldwina Tita Olirk.

**Investigation:** Baldwina Tita Olirk, Aiwerasia Vera Ngowi, Ezra Jonathan Mrema, Jovine Bachwenkizi, Simon Henry Mamuya.

**Methodology:** Baldwina Tita Olirk, Furaha August, Ezra Jonathan Mrema, Jovine Bachwenkizi, Simon Henry Mamuya.

**Project administration:** Baldwina Tita Olirk.

**Resources:** Baldwina Tita Olirk.

**Software:** Baldwina Tita Olirk.

**Supervision:** Baldwina Tita Olirk, Aiwerasia Vera Ngowi, Furaha August, Ezra Jonathan Mrema, Simon Henry Mamuya.

**Validation:** Baldwina Tita Olirk.

**Visualization:** Baldwina Tita Olirk.

**Writing – original draft:** Baldwina Tita Olirk, Ezra Jonathan Mrema, Jovine Bachwenkizi, Simon Henry Mamuya.

**Writing – review & editing:** Baldwina Tita Olirk, Aiwerasia Vera Ngowi, Furaha August, Ezra Jonathan Mrema, Jovine Bachwenkizi.

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
