## [Decision Letter · Decision Letter 0]

17 Dec 2024

PONE-D-24-38857Maternal occupation and risk of adverse fetal outcomes in Tanzania: a Hospital- based cross-sectional studyPLOS ONE

Dear Dr. Olirk,

Thank you for submitting your manuscript to PLOS ONE. After careful consideration, we feel that it has merit but does not fully meet PLOS ONE’s publication criteria as it currently stands. Therefore, we invite you to submit a revised version of the manuscript that addresses the points raised during the review process.

Please submit your revised manuscript by Jan 31 2025 11:59PM. If you will need more time than this to complete your revisions, please reply to this message or contact the journal office at plosone@plos.org . Please include the following items when submitting your revised manuscript:

We look forward to receiving your revised manuscript.

Kind regards,

Kahsu Gebrekidan, Ph.D.

Academic Editor

PLOS ONE

Journal Requirements:

“Funding was provided by SIDA project from Muhimbili University of Health and Allied Sciences”

3. Please note that funding information should not appear in the Acknowledgments section or other areas of your manuscript. We will only publish funding information present in the Funding Statement section of the online submission form. Please remove any funding-related text from the manuscript. 

4. We note that there is identifying data in the Supporting Information file <S1 Data. The data set used for analysis.xlsx>. Due to the inclusion of these potentially identifying data, we have removed this file from your file inventory. Prior to sharing human research participant data, authors should consult with an ethics committee to ensure data are shared in accordance with participant consent and all applicable local laws.

-Location data

Please remove or anonymize all personal information (Age), ensure that the data shared are in accordance with participant consent, and re-upload a fully anonymized data set. Please note that spreadsheet columns with personal information must be removed and not hidden as all hidden columns will appear in the published file.

Reviewers' comments:

Reviewer's Responses to Questions

**Comments to the Author**

1. Is the manuscript technically sound, and do the data support the conclusions?

Reviewer #1: Partly

Reviewer #2: Yes

Reviewer #3: Partly

2. Has the statistical analysis been performed appropriately and rigorously? 

Reviewer #1: Yes

Reviewer #2: Yes

Reviewer #3: Yes

3. Have the authors made all data underlying the findings in their manuscript fully available?

Reviewer #1: Yes

Reviewer #2: Yes

Reviewer #3: Yes

4. Is the manuscript presented in an intelligible fashion and written in standard English?

Reviewer #1: No

Reviewer #2: Yes

Reviewer #3: No

5. Review Comments to the Author

Reviewer #1: In the abstract, the necessity of the research is not clearly stated.

At the end of the conclusion, make suggestions for future studies.

State the strengths of your study at the end of the discussion.

Reviewer #2: Thank you for this study.

It is a good addition to support the impact of agricultural chemicals on the fetal outcomes.

The study was well written, and data were adequate enough to support the conclusion of your study.

Please find below my comments:

Line 20: Consider replacing the phrase with "reproductive health"

Line 66-68: Reference please

Line 81-83: This sentence seems to break the good flow of your social argument of the study.

I do not think it is necessary.

Rather elaborate on the strength of the contradictory findings. Thanks

Line 83: I think it is worth mentioning that the women in labor force is actually highest in the sub-saharan African compared to any other regions globally.

Line 84: You mean " pregnancy"?

Line 93: This is a good description of the study setting.

Thank you.

Line 118- 119: Can you explain why this was considered a good criterion when some studies actually support the increase incidence of preterm delivery in multigravida women exposed to frequent excessive occupational pressures?

Line 118-120: is preeclampsia or Gestational DM exclusion? if so why/

Why is HIV considered an exclusion?

Line 136- 137: Very important denominator in defining the risk associated with the undesirable outcomes if any.

Line 184: I think the obstetric high risk age groups are supposed to be independently categorized... < 17 and >35.

The outcome variables in these group may not strictly be due to the occupational exposures

Consider an adjustment in this regard.

Line 209: I think your discussion of the data above were too precise with little or no diversification from the previous studies.

This will affect both the social and scientific value of the study.

I will suggest you go through the data again and identify some more unique characteristic of this study compared to similar studies over the world.

Thanks.

Line 217- 219: This finding is in line with the data in your study, however, I am expecting that your discussion will acknowledge the following:

1. Farming and Agriculture still remains the commonest occupation in sub–Saharan Africa.

2. Farming and Agriculture in Tanzania may not be as mechanized as in the California where the quoted research was conducted.

please consider how these two factors could affect the validity of your study outcomes

Line 250-254: I think there are more limitations to this study.

It is well researched but the limitations are so many e.g. Most women in sub saharan African are actually multitasking to keep their home .. most of the people documented as working in agriculture sector could still be street food vendors. This study didn't assess the cumulative effect of this.

Please identify others.

Good luck

Reviewer #3: The topic and objective of this study address one of the most pressing health issues in developing and underdeveloped societies. Its findings could provide a basis for improving women’s employment conditions, labor laws, and occupational and environmental health policies. Enhancing the methodology, resolving ambiguities, and expanding the discussion would further strengthen the study's impact. Additionally, the text requires grammatical corrections and linguistic refinement to improve its overall clarity and quality.

Comments:

Line 117: Purposive sampling, while useful for targeted populations, limits the generalizability of the findings due to potential selection bias. Random sampling would have provided stronger external validity.

Line 117: While an 80% response rate is reasonable, it still leaves room for non-response bias. The study does not discuss characteristics of non-respondents, which could help assess this bias. Analyze demographic and clinical characteristics of non-respondents to understand potential biases in the data.

Line 118: Collecting data exclusively within 72 hours of delivery may exclude women who give birth outside healthcare facilities, potentially introducing systematic bias by underrepresenting specific populations, such as those in rural or underserved areas. Expanding the data collection timeframe and including a wider range of delivery settings, such as home births, could help ensure a more representative sample.

Line 120: What criteria were used to diagnose heart disease, kidney disease, sickle cell anemia, and HIV? Were these conditions assessed using a standardized method consistently applied to all study participants by the researchers during data collection, or were they documented based on previous medical records with varying approaches? Clarification is needed.

Line 127: Occupational histories and physical demands rely on maternal recall, which could be subject to recall bias, especially in a retrospective framework. Supplement self-reported occupational data with workplace inspections or supervisor verification to reduce recall bias

Line 140: The study assesses adverse outcomes only immediately after delivery, missing longer-term neonatal or maternal health impacts.

Line 143: Although the questionnaire was translated into Swahili, nuances in occupational terms or descriptions may not have been perfectly captured, potentially impacting the accuracy of the exposure data.

Line 211: "The commonest type of occupation was agriculture, tailoring and hairdressers." should use plural verb agreement

Line 214: "The type of occupation positively associated with the largest number of adverse fetal outcome was agriculture." Correct usage: "outcomes were agriculture."

Line 216: "The higher odds of adverse fetal outcomes among agricultural female workers is related to pesticide exposure." Correct usage: "are related to pesticide exposure."

Line 249: Provide a brief discussion of possible interventions (e.g., protective measures or policy recommendations) to mitigate occupational exposures for pregnant women.

Line 250: The limitation section is underdeveloped. Include details on how biases (e.g., recall, reporting) and confounding variables (e.g., socioeconomic factors) might have affected results.

6. PLOS authors have the option to publish the peer review history of their article (what does this mean? ). If published, this will include your full peer review and any attached files.

**Do you want your identity to be public for this peer review?** For information about this choice, including consent withdrawal, please see our Privacy Policy .

Reviewer #1: No

Reviewer #2: **Yes: ** Prof. Adeloye Amoo Adeniji (MBBS; MMed; FCFP; FACRRM).

Reviewer #3: No

---

## [Author Response · Author response to Decision Letter 1]

25 Jan 2025

Table 1: Response to Academic Editor and Reviewers Comments

S/N COMMENT RESPONSE

ACADEMIC EDITOR COMMENTS

https://journals.plos.org/plosone/s/file?id=ba62/PLOSOne_formatting_sample_title_authors_affiliations.pdf Thank you for the reminder; we have revised the manuscript based on the PLOS ONE requirements.

2 Thank you for stating the following financial disclosure: “Funding was provided by SIDA project from Muhimbili University of Health and Allied Sciences” Please state what role the funders took in the study. If the funders had no role, please state: "The funders had no role in study design, data collection and analysis, decision to publish, or preparation of the manuscript." If this statement is not correct you must amend it as needed. Please include this amended Role of Funder statement in your cover letter; we will change the online submission form on your behalf. Thank you for this information. We have amended this part as recommended in the cover letter

3 Please note that funding information should not appear in the Acknowledgments section or other areas of your manuscript. We will only publish funding information present in the Funding Statement section of the online submission form. Please remove any funding-related text from the manuscript. Thank you for this observation. We have removed the funding information from the manuscript

4 We note that there is identifying data in the Supporting Information file <S1 Data. The data set used for analysis.xlsx>. Due to the inclusion of these potentially identifying data, we have removed this file from your file inventory. Prior to sharing human research participant data, authors should consult with an ethics committee to ensure data are shared in accordance with participant consent and all applicable local laws.

-Location data

Additional guidance on preparing raw data for publication can be found in our Data Policy

(https://journals.plos.org/plosone/s/data-availability#loc-human-research-participant-data-and-other-sensitive-data) and in the following article: http://www.bmj.com/content/340/bmj.c181.long.

Please remove or anonymize all personal information (Age), ensure that the data shared are in accordance with participant consent, and re-upload a fully anonymized data set. Please note that spreadsheet columns with personal information must be removed and not hidden as all hidden columns will appear in the published file..

Thank you for this observation. We have removed personal information (Age) from the file as recommended

REVIEWER ONE

1 In the abstract, the necessity of the research is not clearly stated.

Thank you

The necessity has been added: Line 19-21

2 At the end of the conclusion, make suggestions for future studies.

Thank you for this good observation

We have added some few lines (345-351) at the end of the conclusion as recommended

3 State the strengths of your study at the end of the discussion Thank you for this good observation

We have added this part at the end of the discussion that appears from line 308-315

REVIEWER TWO

1. Thank you for this study.

It is a good addition to support the impact of agricultural chemicals on the fetal outcomes.

The study was well written, and data were adequate enough to support the conclusion of your study. Thank you.

2 Line 20: Consider replacing the phrase with "reproductive health" Thank you for the suggestion.

Revision has been made accordingly

3 Line 66-68: Reference please Thank you for the observation We have added a reference

4 Line 81-83: This sentence seems to break the good flow of your social argument of the study.

I do not think it is necessary.

Rather elaborate on the strength of the contradictory findings. Thanks Thank you for the suggestion. We have removed the sentence and added a few new lines (85-86) to support our argument

5 Line 83: I think it is worth mentioning that the women in labor force is actually highest in the sub-saharan African compared to any other regions globally.

Thank you for the suggestion.

We have added this sentence as recommended in line 81-84

6 Line 84: You mean " pregnancy"? Thank you.

We have replaced “maternal period” with “pregnancy” as recommended

7 Line 93: This is a good description of the study setting.

Thank you. Thank you

8 Line 118- 119: Can you explain why this was considered a good criterion when some studies actually support the increase incidence of preterm delivery in multigravida women exposed to frequent excessive occupational pressures? Thank you for this good observation.

The point of multigravida is valid and that this group of women might also face certain.

We would like to clarify that we did not exclude multigravida but we excluded multiple pregnancies (e.g., twins) from our study. Therefore, all single-tone pregnancies, regardless of gravidity, were included in the study

The reasons why our study focused on singleton pregnancies are reducing variability (maintaining the homogeneity) and minimizing confounding factors. our focus on singleton pregnancies aimed to provide more consistent results related to maternal occupational health risks

We have added some few lines (121-123) to state the reasons for excluding multiple pregnancy

9 Line 118-120: is preeclampsia or Gestational DM exclusion? if so why/ Gestational hypertension, gestational diabetes, preeclampsia, and eclampsia were not excluded from the study. This information was collected and later controlled for in the logistic regression analysis presented in Table 4. The final section of the table is labeled 'Controlled for age, gestational hypertension, gestational diabetes, preeclampsia, and eclampsia.'"

10 Why is HIV considered an exclusion? HIV was excluded due to its potential independent effects on pregnancy outcomes, such as low birth weight and preterm birth. This exclusion also aimed to isolate the impact of occupational factors on pregnancy outcomes, ensuring a clearer analysis of occupational health risks.

11 Line 136- 137: Very important denominator in defining the risk associated with the undesirable outcomes if any. Thank you

12 Line 184: I think the obstetric high risk age groups are supposed to be independently categorized... < 17 and >35.

The outcome variables in these group may not strictly be due to the occupational exposures

Consider an adjustment in this regard. Thank you.

We have revised the age Categories in table 1 as recommended.

Also, age in categories has been adjusted for in the logistic regression analysis presented in Table 4.

13 Line 209: I think your discussion of the data above were too precise with little or no diversification from the previous studies.

This will affect both the social and scientific value of the study.

I will suggest you go through the data again and identify some more unique characteristic of this study compared to similar studies over the world.

Thanks. Thank you for this observation.

We have revisited our data and made some additional analysis on prevalence of adverse birth outcome as per occupational category presented in Table 3 and added discussions under these results to present some unique findings in our study while comparing the findings from other similar studies

We have added some lines (223-258; 262-271; 292-295) in the discussion

14 Line 217- 219: This finding is in line with the data in your study, however, I am expecting that your discussion will acknowledge the following:

1. Farming and Agriculture still remains the commonest occupation in sub–Saharan Africa.

2. Farming and Agriculture in Tanzania may not be as mechanized as in the California where the quoted research was conducted.

please consider how these two factors could affect the validity of your study outcomes Thank you for this suggestion

We have made the changes as recommended (line 268-271)

15 Line 250-254: I think there are more limitations to this study.

It is well researched but the limitations are so many e.g. Most women in sub saharan African are actually multitasking to keep their home .. most of the people documented as working in agriculture sector could still be street food vendors. This study didn't assess the cumulative effect of this. Please identify others. Thank you for this observation and suggestion

We have made the changes as recommended in line 332-334

16 Good luck Thank you

REVIEWER THREE

1 The topic and objective of this study address one of the most pressing health issues in developing and underdeveloped societies. Its findings could provide a basis for improving women’s employment conditions, labor laws, and occupational and environmental health policies. Enhancing the methodology, resolving ambiguities, and expanding the discussion would further strengthen the study's impact. Additionally, the text requires grammatical corrections and linguistic refinement to improve its overall clarity and quality.’ Thank you for your insightful and constructive comments.

We greatly appreciate your recognition of the study's relevance to improving women's employment conditions and occupational health in developing and underdeveloped societies.

Your suggestions to enhance the methodology, clarify ambiguities, and expand the discussion have been valuable, and we have made the necessary revisions to address these points.

We also have worked on the grammatical errors and have carefully refined the text to improve clarity and quality.

2 Line 117: Purposive sampling, while useful for targeted populations, limits the generalizability of the findings due to potential selection bias. Random sampling would have provided stronger external validity. Thank you for this observation

We acknowledge that purposive sampling limits the generalizability of our findings due to potential selection bias. However, this method targeted specific populations critical to the study objectives.

We have added this point as a limitation in the manuscript in line 316-317 and recommended future research to employ random sampling to enhance external validity and validate these findings across broader populations in line 345-346

2 Line 117: While an 80% response rate is reasonable, it still leaves room for non-response bias. The study does not discuss characteristics of non-respondents, which could help assess this bias. Analyze demographic and clinical characteristics of non-respondents to understand potential biases in the data. Thank you for this observation.

We acknowledge that non-response bias is a potential limitation of this study, despite the reasonable 80% response rate.

Unfortunately, data on the characteristics of non-respondents were not collected, which limits our ability to assess this bias directly.

However, to account for the design effect, we adjusted the sample size by multiplying it by 1.5, which, combined with a high response rate, minimized the likelihood of significant bias.

We have added some texts in line 327-329 in the limitations

3 Line 118: Collecting data exclusively within 72 hours of delivery may exclude women who give birth outside healthcare facilities, potentially introducing systematic bias by underrepresenting specific populations, such as those in rural or underserved areas. Expanding the data collection timeframe and including a wider range of delivery settings, such as home births, could help ensure a more representative sample. Thank you for this comment

We acknowledge that collecting data exclusively within 72 hours of delivery in healthcare facilities may exclude women who give birth outside these settings, potentially introducing systematic bias.

However, The data collection area is situated in urban regions, where the likelihood of women delivering at home or coming from rural areas is low. Most women in urban settings prefer to deliver at the hospital

We also chose to collect data within 72 hours to minimize recall bias especially on the questions regarding occupational exposure during pregnancy.

We have added some texts in the limitation. Please see line 329-332

we also recommend future studies (line 347-348) to address to expand the data collection timeframe and including home births to capture a more representative sample

4 Line 120: What criteria were used to diagnose heart disease, kidney disease, sickle cell anemia, and HIV? Were these conditions assessed using a standardized method consistently applied to all study participants by the researchers during data collection, or were they documented based on previous medical records with varying approaches? Clarification is needed. We appreciate this observation

We have clarified in the manuscript that the diagnosis of heart disease, kidney disease, sickle cell anemia, and HIV was based on documented medical records provided by the Hospital (line 124-125).

5 Line 127: Occupational histories and physical demands rely on maternal recall, which could be subject to recall bias, especially in a retrospective framework. Supplement self-reported occupational data with workplace inspections or supervisor verification to reduce recall bias Thank you for this observation

We acknowledge that relying on maternal recall for occupational histories and physical demands may introduce recall bias, especially within a retrospective study design. Unfortunately, we were unable to obtain self-reported occupational data corroborated through workplace inspections or supervisor verification. The majority of study participants were employed in informal sectors such as agriculture, food vending, tailoring, hairdressing, and cosmetology, where occupational health services, including workplace inspections or supervisor verification, are typically unavailable

6 Line 140: The study assesses adverse outcomes only immediately after delivery, missing longer-term neonatal or maternal health impacts Thank you for this observation

We acknowledge that this study focuses solely on adverse outcomes immediately after delivery, which limits insights into longer-term neonatal and maternal health impacts.

This approach was chosen to align with the scope and study's primary objective of assessing short-term health outcomes within the delivery context.

We recommend future research to address this gap by incorporating follow-up assessments at key intervals (e.g., six months or one year postpartum) to better understand the extended trajectory of neonatal and maternal health. Pleases see line 348-350

7 Line 143: Although the questionnaire was translated into Swahili, nuances in occupational terms or descriptions may not have been perfectly captured, potentially impacting the accuracy of the exposure data. We acknowledge that translating the questionnaire into Swahili may not have fully captured cultural or regional variations in occupational terminology, potentially impacting the quality of the exposure data. However, to mitigate this, the Swahili version of the questionnaire was developed in consultation with occupational health experts who are native Swahili speakers. Their expertise ensured that t

---

## [Decision Letter · Decision Letter 1]

6 Feb 2025

Maternal occupation and risk of adverse fetal outcomes in Tanzania: a Hospital- based cross-sectional study

PONE-D-24-38857R1

Dear Dr. Baldwina,

We’re pleased to inform you that your manuscript has been judged scientifically suitable for publication and will be formally accepted for publication once it meets all outstanding technical requirements.

Kind regards,

Kahsu Gebrekidan, Ph.D.

Academic Editor

PLOS ONE

Additional Editor Comments (optional):

Reviewers' comments:

Reviewer's Responses to Questions

**Comments to the Author**

1. If the authors have adequately addressed your comments raised in a previous round of review and you feel that this manuscript is now acceptable for publication, you may indicate that here to bypass the “Comments to the Author” section, enter your conflict of interest statement in the “Confidential to Editor” section, and submit your "Accept" recommendation.

Reviewer #2: All comments have been addressed

Reviewer #3: All comments have been addressed

2. Is the manuscript technically sound, and do the data support the conclusions?

Reviewer #2: Yes

Reviewer #3: Yes

3. Has the statistical analysis been performed appropriately and rigorously? 

Reviewer #2: Yes

Reviewer #3: Yes

4. Have the authors made all data underlying the findings in their manuscript fully available?

Reviewer #2: Yes

Reviewer #3: Yes

5. Is the manuscript presented in an intelligible fashion and written in standard English?

Reviewer #2: Yes

Reviewer #3: Yes

6. Review Comments to the Author

Reviewer #2: Dear Author,

Thank you for appropriately addressing my concerns raised in the previous review of this article.

I think the study is now suitable for publication.

The conduct of the study agrees with the minimum standard required of a cross-sectional study.

The results are well presented intelligently.

Your discussions should provoke scientific arguments in among scholars.

Thank you for adding to the body of knowledge on maternal occupations and fetal outcomes.

Congratulations.

Reviewer #3: (No Response)

7. PLOS authors have the option to publish the peer review history of their article (what does this mean? ). If published, this will include your full peer review and any attached files.

**Do you want your identity to be public for this peer review?** For information about this choice, including consent withdrawal, please see our Privacy Policy .

Reviewer #2: **Yes: ** Adeloye Amoo Adeniji (MBBS; MMED; FCFP; FACRRM)

Reviewer #3: No

---

## [Editor Report · Acceptance letter]

PONE-D-24-38857R1

PLOS ONE

Dear Dr. Olirk,

I'm pleased to inform you that your manuscript has been deemed suitable for publication in PLOS ONE. Congratulations! Your manuscript is now being handed over to our production team.

Kind regards,

on behalf of

Dr. Kahsu Gebrekidan

Academic Editor

PLOS ONE